# The Evolution of Agrarian Landscapes in the Tropical Andes

**DOI:** 10.3390/plants13071019

**Published:** 2024-04-03

**Authors:** Courtney R. Shadik, Mark B. Bush, Bryan G. Valencia, Angela Rozas-Davila, Daniel Plekhov, Robert D. Breininger, Claire Davin, Lindsay Benko, Larry C. Peterson, Parker VanValkenburgh

**Affiliations:** 1Institute of Global Ecology, Florida Institute of Technology, Melbourne, FL 32901, USAangelrodav@gmail.com (A.R.-D.); lbenko2018@my.fit.edu (L.B.); 2Facultad de Ciencias de La Tierra y Agua, Universidad Regional Amazónica Ikiam, Tena 150150, Ecuador; bryan.valencia@ikiam.edu.ec; 3Department of Anthropology, Portland State University, Portland, OR 97201, USA; dan.plekhov@gmail.com; 4Department of Mathematics, University of Richmond, Richmond, VA 23173, USA; cdavin@claritybenefitsolutions.com; 5Rosenstiel School of Marine, Atmospheric, and Earth Science, University of Miami, Miami, FL 33149, USA; lpeterson@miami.edu; 6Department of Anthropology, Brown University, Providence, RI 02915, USA; parker_vanvalkenburgh@brown.edu

**Keywords:** climate change, fossil pollen, lake sediment, terracing, crops, pastoralism, *Sporormiella*, charcoal, XRF

## Abstract

Changes in land-use practices have been a central element of human adaptation to Holocene climate change. Many practices that result in the short-term stabilization of socio-natural systems, however, have longer-term, unanticipated consequences that present cascading challenges for human subsistence strategies and opportunities for subsequent adaptations. Investigating complex sequences of interaction between climate change and human land-use in the past—rather than short-term causes and effects—is therefore essential for understanding processes of adaptation and change, but this approach has been stymied by a lack of suitably-scaled paleoecological data. Through a high-resolution paleoecological analysis, we provide a 7000-year history of changing climate and land management around Lake Acopia in the Andes of southern Peru. We identify evidence of the onset of pastoralism, maize cultivation, and possibly cultivation of quinoa and potatoes to form a complex agrarian landscape by c. 4300 years ago. Cumulative interactive climate-cultivation effects resulting in erosion ended abruptly c. 2300 years ago. After this time, reduced sedimentation rates are attributed to the construction and use of agricultural terraces within the catchment of the lake. These results provide new insights into the role of humans in the manufacture of Andean landscapes and the incremental, adaptive processes through which land-use practices take shape.

## 1. Introduction

The tropical Andes are both a biodiversity and a conservation hotspot, as defined on the basis of high plant endemicity and substantial habitat loss [1,2]. Tropical Andean landscapes have a deep history of human modification as revealed through studies of fossil pollen and charcoal [3,4], spores [5,6], and phytoliths [7,8,9], coupled with archaeology [10,11,12]. Such studies have provided evidence of early human impacts such as the extirpation of megafauna and the introduction of frequent fires [13]. Consequently, the modern landscape, especially above tree-line, is widely regarded as a landscape manufactured by past human actions [14,15,16].

### 1.1. An Historical Overview of Early Settlement and Cultivation

The earliest archaeological evidence of people occupying the high Andes of Peru comes from open-air workshops where stone tools were fashioned [17]. These sites are dated to about 12,500–11,500 calibrated radiocarbon years before present (hereafter cal BP). The initial settlers of the Andes were hunter-gatherers, but archaeological data point to a transition toward more sedentary lifestyles about 8000 cal BP [18]. Coastal indigenous societies had cultivated domesticated squash and beans by 10,000 cal BP [19], but Andean cultivation of crops, such as maize, potato and quinoa, appears to have expanded about c. 7000 cal BP [20,21]. Domestication of livestock may have gone hand-in-hand with crop cultivation. In southern Peru, llama and alpaca domestication may have begun as early as 7000 cal BP [22], possibly triggering a synergy between the corralling of camelids and the cultivation of weedy amaranths (pseudo-cereals such as quinoa). These plants may have been favored by disturbed ground and the high nutrient availability in soils enhanced with dung in and around corrals [23]. By about 4000 cal BP, reliance on domesticated camelids and crops such as *Zea mays* (maize) and *Chenopodium quinoa* (quinoa) had become widespread [21,24]. While we have a basic timeline of the introduction of these practices, there is no single record that shows their development and the ecological consequences of that land use.

### 1.2. Human Responses to Holocene Climate Change and the Temporal Scale of Study

Within this timeframe of plant and animal domestication, frequent and intense droughts associated with the Mid-Holocene Dry Event (MHDE) of c. 9000–4000 cal BP induced the lowest lake levels of the last 100,000 years [25] and probably impacted many aspects of human life [26,27]. Between c. 4000 years ago and the European invasion of the 1500s, many societies made investments in extensive irrigation and drainage canals, raised fields and terraces to regulate water supplies and safeguard soils [28,29,30]. The respective roles of climate change and social factors in driving humans to invest in agriculture and modify landscapes has been actively debated [31,32,33]. Much of this debate has emphasized short-term interactions between humans and their environments and provided mostly monocausal explanations for their resulting changes. Missing from such explanations has been consideration of the influence of antecedent land-use practices and how their legacies create opportunities and obstacles for later land use [34]. To build robust models of these processes, we face the considerable challenge of capturing interactions between social and environmental factors over long timespans (millennia), while also attending to the processes through which human agents make decisions. Creating such models has been complicated for both paleoecologists and archaeologists, as paleoecological studies are often too temporally coarse to capture short-term decision-making, while archaeological research tends to prioritize the study of single periods, missing longer-term trajectories. In addition, land-use practices often fail to leave behind materials that can be easily dated. Where remains (such as terrace walls, corrals, and canal courses) are visible, archaeologists may be able to infer that certain land-use practices occurred but then have trouble dating them due to repeated practices of maintenance and modification, including the re-use of soils and building materials. Thus, in moving from short-term monocausal explanations to more complex models of long-term change, untangling specific relationships of cause and effect becomes more difficult and resulting models risk losing explanatory power [35,36].

Here, we present a 7000-year history that reveals the timing and trajectory of climate and land management around Lake Acopia in the Andes of southern Peru. Specifically, we ask: How did climate change influence land use around Lake Acopia? And can we detect adaptations to environmental change in the way the land was used?

## 2. Site Description, Materials and Methods

Lake Acopia (Figure 1) lies at 3715 m above sea level (hereafter masl), approximately 70 km southeast of Cusco, Peru. This lake lies within the region of the Wari (c. AD 600–1000; 1350–950 cal BP) and Inka (c. AD 1400–1533; 550–417 cal BP) empires and is likely to have been occupied for a much longer period. The pre-Wari history of this region is poorly known, and yet it is considered to lie within the cradle of plant and animal domestication [21]. The climate of this part of the Andes is strongly seasonal with a strong dry season lasting from May–September. The mean annual precipitation in the Cusco area is 1278 mm, with a mean annual temperature of approximately 7.8 °C (https://en.climate-data.org/south-america/peru/cusco/cusco-1016/#climate-graph; accessed 26 February 2024). The basin is bordered by the town of Acopia to the north, potato fields and pastures to the east and south, and a steeper slope to the west that supports *Eucalyptus*—a nonnative tree that has only been commonly cultivated in this landscape during the past 60 years [37]. Although Poaceae are the most abundant cover, and *Eucalyptus* is the commonest large woody species in the basin today, taxa such as *Vallea, Schinus*, and a wide diversity of Asteraceae and Fabaceae would have been important components of the natural vegetation. In 2008, we mapped the bathymetry of Lake Acopia, collected using a hand-held sonar. The lake has a closed basin with a mostly flat bottom, averaging around 20 m deep except for a depression in the west-central portion of the basin that reaches a 98 m depth (Figure 2). As the deepest point in Lake Acopia was beyond the limits of our coring rig, and such a sheer depression would invite slumping, we selected a flat shelf at a depth of 19.5 m for coring (Figure 2a).

A sediment core was raised from the coring point (Figure 2a) using a Colinvaux-Vohnout piston corer from a raft of rubber boats [38]. Eight drives, totaling 7.74 m of recovered sediment, were acquired before hitting a methane pocket in the sediment, halting the coring process. All cores were sealed in the field and sent back to Florida Institute of Technology where they were stored at 4 °C. The split cores were described and one of the two halves subsampled. Eight bulk sediment samples were shipped to Woods Hole Oceanographic Institute NOSAMS (National Ocean Sciences Accelerator Mass Spectrometry) for AMS (accelerator mass spectrometry) dating and one to Direct AMS. The package rbacon version 2.2 [39] was used to create the Acopia age model using the IntCal20 calibration curve [40,41].

Loss-on-ignition at 108 subsampled locations along the core followed the protocol of Heiri et al. [42]. Subsamples were heated in a muffle furnace to 550 °C for 4 h to determine organic content, and at 950 °C for 2 h to quantify carbonate content.

A total of 108, 0.5-cm^3^ sediment subsamples were analyzed for pollen, *Sporormiella*, and maize presence at the same locations as LOI. Pollen samples were prepared using standard palynological procedures [43]. Pollen grains and spores were identified with a Zeiss Axioskop photomicroscope at ×630 magnification using pollen atlases [44,45], the modern pollen collection at Florida Tech and the Neotropical pollen key and database [46]. After initial counts were made, samples were filtered through a 55-µm mesh to facilitate extended counting for large grains [47] such as *Zea mays* (maize). *Zea* was identified as Poaceae grains >80 µm. *Sporormiella* spores were quantified as % of the pollen sum and as concentrations (spores per cm^3^). CONISS software, a multivariate analysis tool to determine significant changes in a time-series, was run on all pollen data [48] to define the local pollen zones.

Sediment subsamples of 0.5 cm^3^ for charcoal analysis were taken continuously every centimeter along the core. The subsamples were filtered with water through a 180-µm mesh to separate macrocharcoal from other material within the sediment. The particles on the mesh were transferred to a petri dish for identification and quantification. All charcoal was identified at ×20 magnification on an Olympus stereoscope, and surface area was quantified using ImageJ version1.54a software [49].

The archive half of the core was scanned with an Avaatech (Dodewaard, The Netherlands) XRF core scanner [50] at the University of Miami, Rosenstiel School of Marine, Atmospheric, and Earth Science. Each refrigerated core was brought to room temperature before analysis to reduce condensation on the sediment surface. The core surface was then gently scraped with a glass slide, to present a fresh surface for scanning. Each core was covered with 0.4-µm-thick Ultralene^®^ to prevent contact between the XRF detector and the sediment surface. The XRF detector read elemental composition at 10 kV and 30 kV at a current of 1000 µA for 20 s every 2 mm along the core.

Major changes in mean erosion, using the Ti record, were determined using the “changepoint” package in R version 2.2.2 with the BinSeg method [51]. The number of meaningful change points was determined from the diagnostic plot created in the “changepoint” package where the addition of a true change point improves model fit. Changes in erosion patterns, as indicated by the true changepoints, were analyzed in combination with maize presence, charcoal, and *Sporormiella* influx as proxies of human landscape disturbance.

For assessing the density and composition of terraces within the Acopia basin, we employed a combination of extensive field survey and satellite imagery analysis. Field surveys took place during the summer of 2022 and involved targeted visits to several parts of the Acopia basin to characterize and document terrace morphologies and ongoing uses. These visits were paired with intensive digitization of all visible terraces in publicly available satellite imagery.

## 3. Results

The modern landscape has at least 233 km of extant terracing within its approximately 33 km^2^ catchment (Figure 3), the vast majority of which appears to be ancient—based on the construction styles used and the frequency of artifacts found on the surface around the terraces.

Four out of the nine bulk sediment ^14^C dates were accepted and used to create the age model (Table 1; Figure 4). The five dates that were rejected as outliers came from the upper 120 cm of sediment. The resulting age model provides a basal age of c. 7140 (calibrated radiocarbon years before present; hereafter cal BP).

The last reliable ^14^C age is at c. 2590 cal BP, and ages in the upper 120 cm are derived by a simple interpolation between that point and the surface, taken to be 2008 when the lake was cored. The identification of nonnative *Eucalyptus* pollen in the topmost sample (1 cm depth), which only became widespread in the Andes after c. 1960 [37], confirms that the top of the core is modern.

Pollen and spores were well-preserved and 136 pollen types were differentiated. The CONISS analysis produced three statistically significant local pollen zones (ACP–1–3).

ACP–1 (768–543 cm; c. 7100–4450 cal BP)

Sediments at the base of the zone were composed of clays with sand rich in carbonates and Ca (Figure 5 and Figure 6). These sediments gave way to dark brown, organic-rich (~30% carbon) laminated gyttja at c. 6380 cal BP (Figure 5). The XRF data for S approximated that of C derived from the LOI analysis, peaking at c. 5580 cal BP before gradually decreasing. The elemental analysis (Figure 6) revealed very similar profiles for the terrigenous elements: AL, Si, K, Ti, Fe, Co, Cu, and Zn (hereafter terrigenous elements). These elements all increased in concentration abruptly c. 5350 cal BP. A further transition to less-carbon rich gyttjas intercalated with tan clays occurred near the top of the zone as sedimentation rates increased c. 4700 cal BP (Figure 5). The pollen concentrations increased throughout this zone, reaching a peak of c. 300,000 grains per cm^3^ c. 4500 cal BP (Figure 7). Poaceae (c. 70%) dominated the pollen spectra with *Plantago,* Amaranthaceae, *Ambrosia,* and other Asteraceae as the next most abundant types. *Zea* was not found, and *Sporormiella* spores were rare. Charcoal was abundant throughout much of this zone. The samples from ACP–1 had predominantly negative values on both Axis 1 and 2 of the DCA (Figure 8).

ACP–2 (542–139 cm; c. 4450–2090 cal BP)

Sediments in ACP–2 were all finely laminated clays and gyttjas with low (~10%) carbon content (Figure 5). The rate of sedimentation increased markedly c. 3200 cal BP, but then declined to near its lowest level of the record at c. 2300 cal BP. The concentrations of terrigenous elements were volatile but showed a marked temporary decline at c. 2300 cal BP (Figure 6). The pollen spectra were broadly similar to those of the preceding zone except for higher abundances of Amaranthaceae, Asteraceae, and *Alnus,* with peaks of c. 41, 22%, and 10%, respectively (Figure 7). Samples from ACP–2 were largely segregated from all other samples by having negative values on DCA Axis 1 and positive values on Axis 2 (Figure 8) due to *Zea* being first recorded at c. 4400 cal BP, with no further record until c. 3700 cal BP, whereafter it was found regularly (Figure 7). *Sporormiella* was also consistently present at above 2% and reached a peak of c. 31% at 3640 cal BP. Three major spikes of fire activity occurred within the zone but, overall, charcoal was rarer than in ACP–1.

ACP–3 (138–0 cm; c. 2090–0 cal BP)

The gross morphology of sediments in ACP-3 were similar to those of ACP–2, although laminations were not as well developed (Figure 5). Sedimentation rates could not be measured in this zone but are inferred to have remained low based on a linear rate of sedimentation from the last reliable ^14^C age to the surface. The terrigenous elements show increasing concentrations until c. 1000 cal BP, before plateauing and being less variable than in ACP–2 (Figure 6). Poaceae and Amaranthaceae pollen continued to decline in abundance as *Alnus*, Asteraceae, and Brassicaceae showed increases. *Zea* was found in the basal samples of the zone but was absent between c. 1780 and 260 cal BP (Figure 7). The samples from this zone all had positive scores on DCA Axis 1 and were almost entirely segregated from those of the other zones (Figure 8). *Sporormiella* rose in abundance throughout this zone before showing a sharp spike to its peak abundance (37%) at c. 460 cal BP. Charcoal was present in low and declining quantities throughout this zone.

## 4. Discussion

The Lake Acopia record provides a detailed history of the rise of a complex agrarian system in the high Andes of southern Peru. We did not capture lake formation, but the start of the record, which was in the midst of the Mid-Holocene Dry Event (MHDE), documents a shallow system with substantial deposition of CaCO_3_ (Figure 5 and Figure 6). The lake began to deepen after c. 6500 cal BP as the evaporitic influence that led to carbonate deposition weakened. At the onset of this record, a puna grassland is suggested by the dominance of Poaceae and low levels of fire activity. Large herbivores were rare in the landscape, and this was a period between the demise of the megafauna [13] but before the rise of pastoralism, when grazing pressures were relatively light. Indeed, the lack of grazing indicated by the rarity of *Sporormiella* (Figure 7) may account for this basal zone (ACP–1) being segregated from other zones in the DCA results (Figure 8).

### 4.1. How Did Climate Change Influence Land Use around Lake Acopia?

The deepening of the lake at c. 6500 cal BP occurred during the peak of the MHDE [25], and does not stand out as being a significant turning point in other regional records [53,54,55]. Consequently, the transition to more organic sediment may reflect local climate effects or that sufficient clays had accumulated to seal the basin and organic material began to accumulate as a permanently inundated system formed. The long-term trend toward wetter conditions following the MHDE (Figure 9) was consistent across regional lake and cave isotopic records [25,55,56]. In the highly-resolved Acopia XRF data, strong decadal-scale droughts were frequent events prior to 4000 cal BP (Figure 8). Peaks of Ca/Ti (Figure 9b) represented prolonged drought events [57] in which lake evaporation caused carbonate to be deposited, while inwash from terrigenous sources, characterized by Ti, fell markedly. These peaks were also represented in the carbonate deposition measure by loss-on-ignition (Figure 5). Charcoal fragments >180 µm were evident in almost all samples in this record (Figure 9c). As regular fire in the Holocene Andes was so rare before human arrival [14], its regular presence is strongly linked to human activity [13,58]. Consequently, the increased abundance of charcoal strongly suggested a human presence at this location for the last 7140 years. Nevertheless, peaks of charcoal clearly aligned with peaks in Ca/Ti, and it was likely that during times of extreme drought normal burning practices escaped to become larger wildfires. The pattern of fire events at Acopia, which reached a peak between 6000 and 4000 cal BP, was also seen at Lake Pacucha [54] (Figure 1).

*Zea mays* (maize) was first observed at Acopia c. 4400 cal BP (Figure 9d) during one of the dry events. This date is quite early for maize agriculture in southern Peru [59], and it may have been grown to provide corn-beer (*chicha*) rather than as a dietary staple [60]. Poaceae pollen was generally present at >30% of the pollen sum, and Asteraceae was often present at >10% (Figure 7). The ratio of these two types, however, has been used to indicate major changes in moisture availability [61]. At Acopia, the Poaceae/Asteraceae ratio (Figure 9e) tracked the Ca/Ti record, suggesting that droughts induced grassland expansion at the expense of herbs and shrubs.

In another potential indication of crop activity, the pollen of Amaranthaceae showed a marked increase in abundance at c. 4400 cal BP (Figure 9f). Although Amaranthaceae can be weedy herbs of high Andean landscapes and wetlands, the pollen was quinoa-type, i.e., it was morphologically attributable to a subset of Amaranthaceae that include the pseudo-cereal quinoa (*Chenopodium quinoa*). In the modern system, where little quinoa was observed, Amaranthaceae inputs were about 2–4%, but its past abundance between c. 4400 and 2800 cal BP was as high as 40% (Figure 9f). Other pollen records from the Cuzco region of Peru inferred quinoa cultivation with pollen abundances of about 10–20% [54,62,63]. It is very likely that quinoa cultivation peaked between 4300 and 3000 cal BP, but declined after 2800 cal BP; this is a pattern that replicates that of other sites in southern Peru where maize increases in apparent importance c. 2800 cal BP [63]. Another potential crop is represented by Solanaceae. Potato (*Solanum tuberosum*) is a native crop of the high Andes that does not produce much pollen [64], and the grains it does produce are only identifiable at the family level. Finding Solanaceae pollen at >2% of the pollen sum is very rare (Table 2). We tentatively suggest that the increase of Solanaceae pollen above the 0–1% level that characterize the record prior to c. 4400 cal BP (Figure 9g) to levels of 3–4% after 4400 cal BP could be indicative of potato cultivation. 

At c. 4400 cal BP, coincident with maize cultivation and the observed increase of quinoa-type pollen, the abundance of *Sporormiella* spores rises above 2% of the pollen sum for the first time (Figure 9h). *Sporormiella* is an obligate dung fungus, and in this context its spores are an indicator of camelid presence. Prior studies have suggested that when *Sporormiella* rises above 2% of the pollen sum, significant numbers of grazers are locally present [70,71]. That there is no prior record of camelids around the lake, coupled with the onset of maize cultivation, is taken to indicate the introduction of domesticated animals to this landscape. These observations and the increase in quinoa could indicate the synergistic inception of corralling and cultivation envisioned by Kuznar (1993) [23], but perhaps it more likely reflects the arrival of a group already using those techniques in the Acopia basin.

The initial occurrence of both maize and *Sporormiella* occurred in a time of low fire activity and no indication of drought. A drought episode at c. 4200 cal BP, when there was a spike of charcoal as well as other drought indicators, saw declines in all the indicators of agrarian activity (Figure 9c,f). As humid conditions returned at c. 4000 cal BP, *Sporormiella* increased in abundance, and maize pollen was found in most samples (Figure 9d,i). Sustained *Sporormiella* occurrence was negatively related to Ca/Ti values, indicating that pastoralism was sensitive to extreme drought. Quinoa-type pollen reached its peak of 41% at c. 3300 cal BP, coinciding with the only samples where maize pollen grains were sufficiently abundant to be found in the original count of 300 grains rather than only encountered in extended counts. The overall trajectory of vegetation change was documented in the results of a multivariate analysis of all fossil pollen types (but not spores) using detrended correspondence analysis (DCA). Axis 1 scores of the DCA (Figure 9j) showed little consistent pattern until c. 4400 cal BP. Thereafter, the vegetation composition followed a new path, which increasingly diverged from the Mid-Holocene state and appeared to accelerate after c. 2800 cal BP. These vegetational trajectories reflected the intensifying impacts of both cultivation and pastoralism on the system. A pollen type that shows a marked increase at c. 1300 cal BP is Brassicaceae (Figure 9h, Table 2). Brassicaceae pollen cannot be identified below the family level, and they can be weeds of arable land. The pattern of Brassicaceae pollen presence at <2% during phases of maize cultivation and herding provides a baseline for this kind of weedy input. As Brassicaceae pollen is found in multiple samples at >10%, we infer that it was being grown as a crop. Maca (*Lepidium meyeni)* is a root crop in the Brassicaceae that is grown between 3800 and 4300 m asl. Maca provides a protein-rich flour, and a drink that promotes stamina [72] and is reputed to assist in animal and human fertility at high elevations [73]. Pearsall [74] found maca wild-type macroremains in Panaulauca Cave in Central Peru (Figure 1) that dated to c. 3550 cal BP, but modern-sized maca roots were represented by c. 950 cal BP. She also suggested a link to camelid husbandry as it may have been an important source of iodine and iron in the camelid diet [72]. Furthermore, the disturbance and fertilization by camelids might favor maca [74]. Fossil pollen records of maca are scarce, but a spike of Brassicaceae to 30% in a single sample from Marcacocha (Table 2) may have been be linked to maca cultivation [62].

We infer that around 4400 cal BP a complex agrarian system developed in which domesticated camelids were raised in combination with quinoa, maize, and potato cultivation. The initial stages of these innovations appear to have been somewhat susceptible to disruption by drought as they show downturns of representation during droughts signaled by the Ca/Ti ratios (Figure 9). As late Holocene climates stabilized, and the management of the system matured, crop representation became less variable. While each component of this agropastoral system is already attested in other parts of the Andes by this date, the Acopia data provide new insights into the scale at which these different components were being used at this time, as well as their interconnectedness within the local subsistence economy. That these components can be inferred as being intensely exploited together at such an early date demonstrates the longevity of complex agrarian landscapes in the Andes, well before the emergence of complex state societies.

### 4.2. Can We Detect Adaptations to Environmental Change in the Way the Land Was Used?

During transition out of the MHDE, significant changes were taking place in the watershed that foreshadowed the need for adaptation. Between c. 5300 and 2300 cal BP, the sediment chemistry indicated large variance in the abundance of elements associated with erosion. Pulses of erosion were indicated by elevated values of inwash of terrigenous inputs, e.g., Ti, Fe, and Rb/Sr, during wet events, and lower values as runoff reduced during drier ones (Figure 10c–e). Despite the period from c. 2300 cal BP to the modern period being even wetter (Figure 10a) than the period with high rates of erosion [55], the variance in terrestrial inputs disappeared, and deposition slowed to its lowest level of the record (Figure 10b–e).

The time period from c. 2290 cal BP to the modern period stands out in the Acopia data because the mean levels of terrigenous input (Ti) and soil erosion (Rb/Sr) and the overall rate of sediment accretion decreased to values similar to those of c. 5000 cal BP, i.e., before maize agriculture and pastoralism (Figure 10). Thus, despite evidence of intense land use and increasing precipitation, the amount of erosion paralleled that of an earlier time with a less disturbed landscape.

We infer that this pattern of reduced sedimentation at 154 cm depth, i.e., 2 cm above the last reliable ^14^C age, was due to the onset of terracing in the catchment of Lake Acopia. While deposition of *Sporormiella* spores continued after terraces were constructed, the representation of maize pollen was interrupted from c. 1700 cal BP until c. 300 cal BP. Such interruption does not necessarily imply that maize was no longer being cultivated, however. Maize pollen, which because of its size (>80 mm) was most likely carried by surface flow into lakes, may have been trapped by newly constructed terraces before entering the lake. *Sporormiella*, on the other hand, was directly deposited by animals defecating on the shoreline while drinking [70]. Maca pollen may also have been directly defecated by camelids if they were fed flowers and foliage.

Support for terracing may also be seen in the Acopia ^14^C record, which is complicated by reversals in the top 1.2 m (last 2000 years) of sediment (Table 1, Figure 4). Age reversals are not uncommon in series of ^14^C dates from paleo sequences where people have induced landscape erosion [65,75,76]. In these systems, erosion creates an influx of “old carbon” from the watershed, contaminating the signal of ^14^C that accumulates in lake sediments [77,78]. In Amazonia and the Andes, the dating of basement soils––i.e., C horizons––commonly produces ^14^C ages in excess of 15,000 years [79], and thus these subsoils contain a substantial “old carbon” component. The two ages with problematic dates, from samples obtained between 81 and 120 cm at Acopia, are inferred to be in error (i.e., the difference between inferred age and measured age) by c. 1530 and 1500 years respectively, while the two overlying dates from 7 and 71 cm had even larger errors of c. 5650 and 4170 years, respectively (Table 1). Broadly, the scale of the error must reflect the proportion of ‘old’ carbon, or the input of older material. As all four of these problematic dates occur within the period of overall decreased terrigenous input, we offer an explanation based on terrace construction.

After vegetation was cleared to create areas for cultivation or to promote grazing, the soil surface profile was disturbed (Figure 11), and the landscape became more prone to erosion [80]. Erosion then increased the influx of “old carbon” from sediments, which increases in age as deeper soils and clay subsoils are eroded. Earthen terraces were built by digging out a portion of soil from upslope, then using that material to fill the terraced structure below [81]. This process resulted in a sediment profile where red clays from eroded Andean subsoils [82] were deposited with landscape disturbance.

Over time, terracing would generally have stabilized landscapes and reduced erosion, but renewed terrace construction would have produced pulses of erosion within the landscape as they were built. Thus while all age reversals were probably caused by older material being washed into the lake, the erosion patterns shown by the XRF data suggest an additional process impacting the age reversals at 7 and 71 cm. Increased values of Rb/Sr suggest more intense weathering in the landscape associated with clay input, as Rb is mostly associated with fine silts and clays [83,84]. When compared with sediment stratigraphy, red clays began depositing again c. 1100 cal BP, coincident with the increase in Rb/Sr and larger error in the ^14^C ages (Figure 10d). Therefore, the increasing age reversals in the top two ages of the core interpreted with Rb/Sr and red clay deposit could reflect a later significant expansion of terracing around Lake Acopia.

We note that the beginning of this expansion coincides with the peak of the Wari culture [85], which is known to have exploited terracing systems throughout their empire [86]. Terrace expansion at Acopia then reaches a peak c. 550 cal BP during the period of Inkan expansion, again a time of pronounced landscape transformation and extensive terracing [87]. The trough in Rb/Sr at c. 420 cal BP occurs at the time of Spanish conquest [88], and may be the consequence of depopulation [89] and temporary terrace abandonment.

The identification of widescale terracing in the Acopia lake records is significant methodologically and anthropologically. Methodologically, most archaeologically-determined terrace chronologies rely on dating basal soils [90] or the soils beneath terrace walls [91], but the inherently disturbed nature of the setting produces results with very large uncertainties. Moreover, it is challenging to infer the scale of regional terracing from a handful of terrace excavations, making it difficult to evaluate the scale or intensity of land use during any one period. By investigating the environmental consequences of terraces (i.e., pronounced changes in sedimentation), a more representative landscape-wide signal of terracing can be measured and dated. At a broader scale, we have encountered many lakes in the Andes where the sedimentary record exhibited significant ^14^C reversals in the last 2000 years or where the youngest age at the core surface was about 2000 years. We came to know this as the ‘paleo Y2K problem’. The basins were frequently steep-sided and terraced, and we infer that a similar analysis to that reported here would determine the onset of terracing in those systems as well.

As for the anthropological significance of these results, the emergence of widescale terracing around 2300 BP provides strong support for the handful of similarly early published terrace dates. Terrace excavations from the Colca Valley, for example, have reported dates as early as 3445 ± 85 BP [29], while the earliest dates from the Chichas-Soras Valley were reported at 3537 ± 28 BP and 2562 ± 36 [91]. For many such early dates, there is skepticism and uncertainty around whether they actually relate to terrace construction, or even to human activity. Our results lend support to the idea that terracing was practiced in the Andes well before the rise of centralized state societies.

We observe that that the apparent expansion of terracing c. 2300 cal BP coincided with the strongest drought in almost 2000 years. This drought may have played a role in motivating strategies to retain water on cultivated areas and maintain productivity in the face of climatic change. Yet to rest on a monocausal climatic explanation for terracing at Acopia would be to ignore the sequence and history of land-use practices that preceded terracing. Erosion rates were already increasing well before 2300 cal BP as a result of long-term exploitation of the landscape’s hillslopes for crop cultivation and camelid grazing—practices that themselves waxed and waned in intensity with periods of aridity throughout the MHDE. The decision of farmers around Acopia to invest in terraces 2300 years ago was therefore informed as much by the climatic conditions at that time as it was by the landscape and suite of crops, animals, and technologies they had inherited from their ancestors.

## 5. Conclusions

The data from Lake Acopia suggest that ancient people who lived near the lake continually adapted their land-use practices to changing conditions––and that they were driven to do so not by a single factor but by the intersecting effects of climate change and the unforeseen consequences of their own, prior land use strategies. Previous studies have shown that Andean grasslands were already modified by the loss of megafauna and increased fire activity in the early Holocene, before the start of this record. The onset of pastoralism and mixed crop usage at Acopia c. 4400 cal BP was thus a continuation of much longer-term patterns of human landscape modification in the southern Peruvian Andes. Over time, crop cultivation, fire, and trampling caused top soil to become unstable; then, farmers built terracing c. 2300 cal BP to mitigate this process. The acute trigger to initiate the terracing effort may have been a drought, suggesting that local societies were sensitive to short-term climate change, but the ultimate need for soil stabilization was the result of a millennium of elevated erosion. Once present, terracing caused an abrupt decrease in terrigenous elements, including surface-washed pollen entering the lake. Despite the continuous use of the landscape and increasing precipitation after 2300 cal BP, the effects of this major alteration to the watershed have lasted until the present. The appearance of terracing coincides with progressively less accurate ^14^C ages in the upper part of the sediment core, and the generalizable pattern of a rapid reduction in sedimentary rate and intense age reversals near the top of the core should be investigated for further evidence of terracing.

## Figures and Tables

**Figure 1 plants-13-01019-f001:**
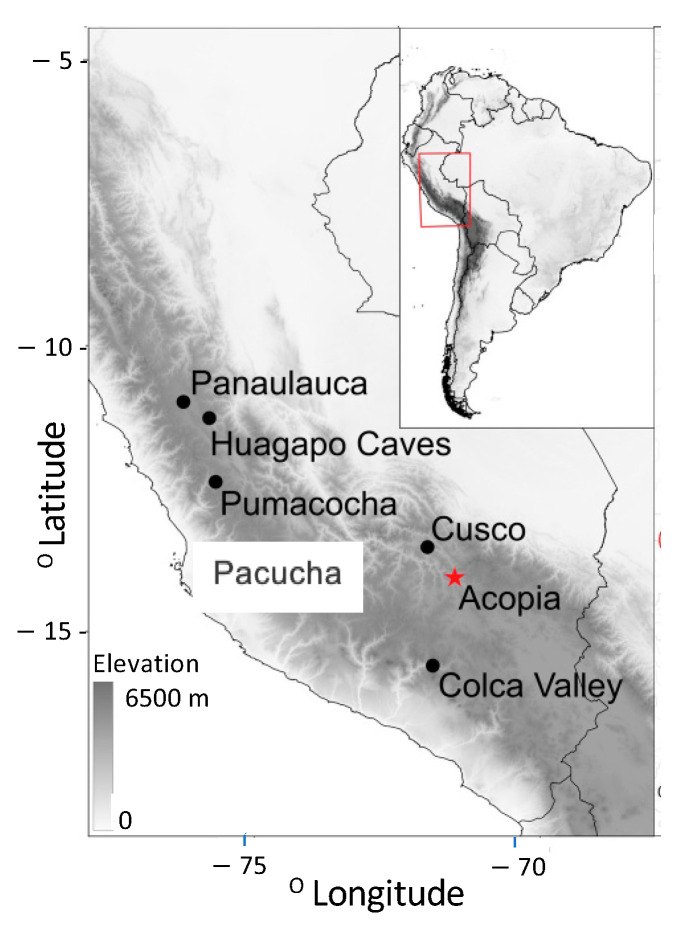
Relative location of Lake Acopia (red star) in relation to the Andes and other sites mentioned in text.

**Figure 2 plants-13-01019-f002:**
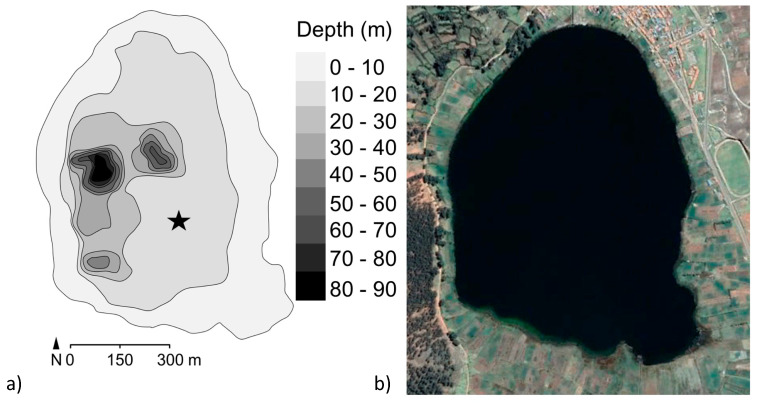
(**a**) The bathymetry of Lake Acopia relative to modern agricultural settings and terracing. The star marks the coring location. (**b**) Google Earth image of Lake Acopia. Trees shown to the left of the lake are the exotic *Eucalyptus*.

**Figure 3 plants-13-01019-f003:**
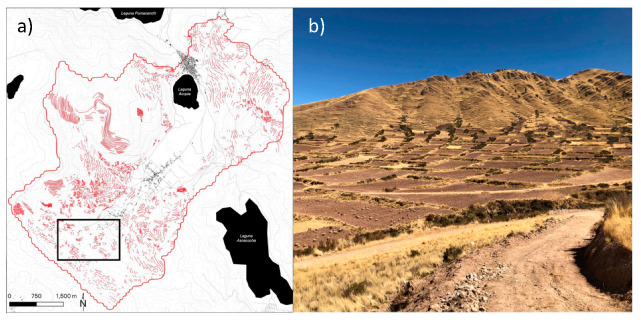
(**a**) Terracing mapped within the watershed of Lake Acopia (derived from Google and Bing imagery), black outline shows approximate location of photograph in Panel (**b**). (**b**) modern terracing beside Lake Acopia, Peru.

**Figure 4 plants-13-01019-f004:**
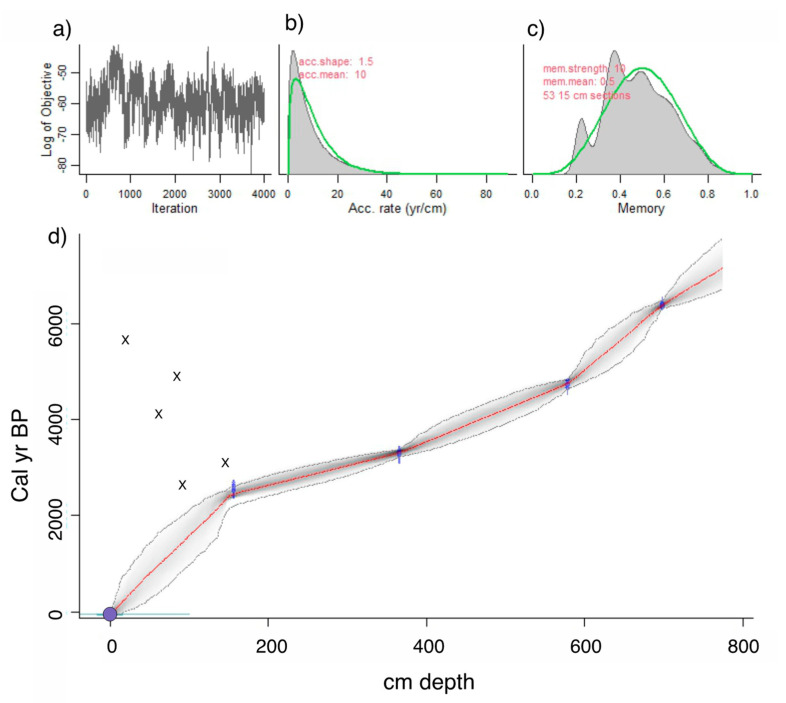
Bayesian chronology of the sediment core raised from Lake Acopia. The age model is developed using rbacon [52] and the IntCal20 calibration curve [40]. Panel (**a**) shows the Markov Chain Monte Carlo iterations with little structure among neighboring iterations (desirable); (**b**) provides the distribution of accumulation rates within the core and (**c**) the memory strength, which determines how much influence preceding samples have on the interpolated curve. In b and c, the green curves are the priors, and grey histograms are the posterior distributions. (**d**) The age-depth model for Lake Acopia shows calibrated ^14^C dates as transparent blue shapes whose heights display the 1 σ range of the calibrated ages, and widths display the most probable range of calibrated ages used in the age model iterations. Darker greys indicate more likely calendar ages. Grey dotted lines depict 95% confidence intervals. The red dotted line depicts the best model based on the weighted mean age for each depth. The purple circle depicts an age based on an exotic species. ‘X’s mark the calibrated ages of dates rejected as outliers.

**Figure 5 plants-13-01019-f005:**
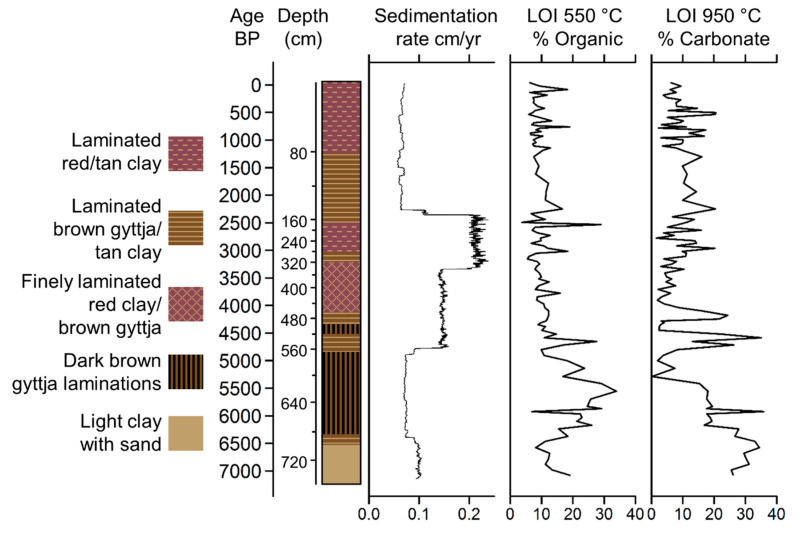
Stratigraphy of the Lake Acopia sediment scaled by age. The right three panels portray sedimentation rates and loss-on-ignition estimates of % organic and % carbonate content.

**Figure 6 plants-13-01019-f006:**
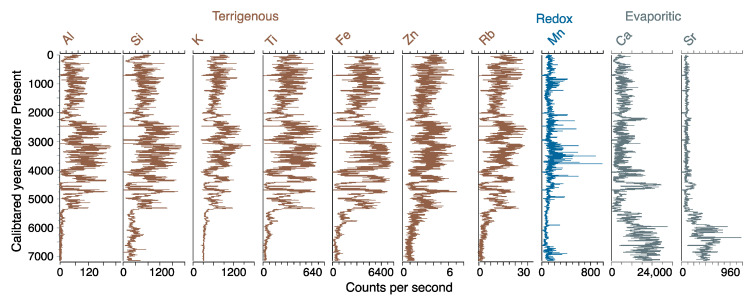
Results of the X-ray fluorescence analysis of selected elements in the sediments of Lake Acopia, Peru.

**Figure 7 plants-13-01019-f007:**
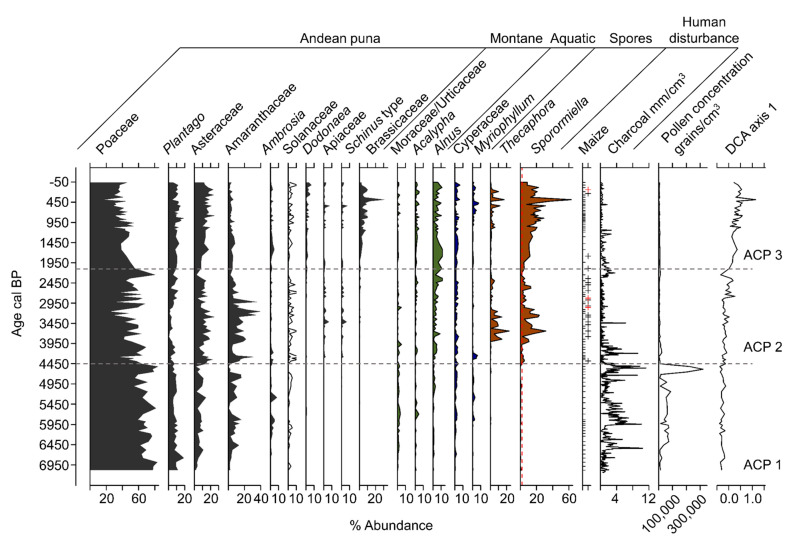
The common fossil pollen types of Lake Acopia. Pollen assemblages through time showing taxa with >2% abundance in five or more samples. Empty lines on plots show ×5 exaggeration. Puna vegetation shown in black, forest (non-local) vegetation shown in green, aquatics shown in blue, and spores shown in brown. The dashed red line on the *Sporormiella* plot marks 2% abundance. Tick marks on maize plot show sampling resolution for extended maize; + indicates maize presence (red + indicates maize found in initial 300 grain pollen count).

**Figure 8 plants-13-01019-f008:**
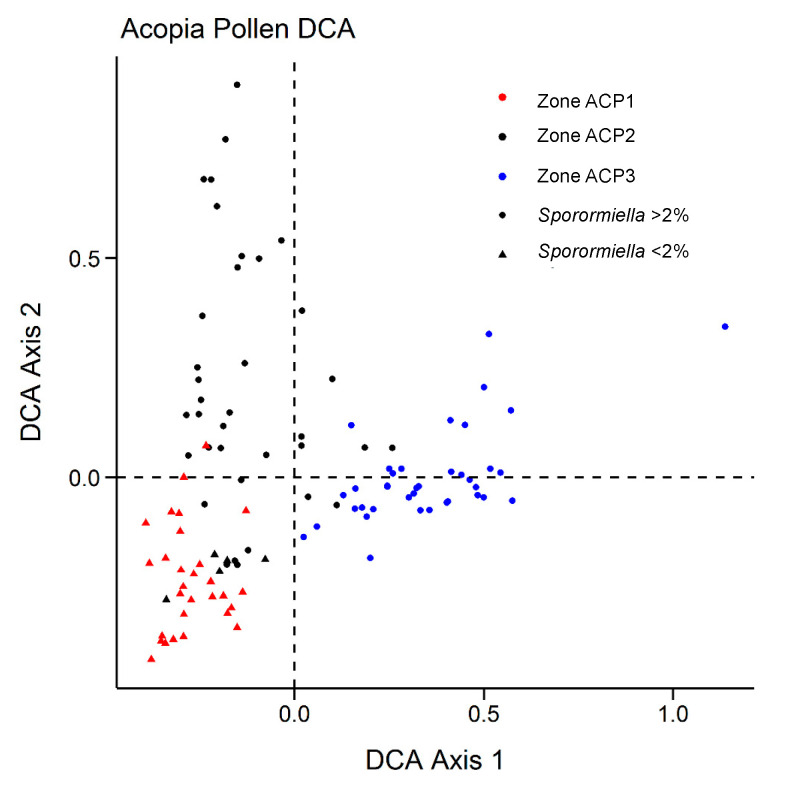
Results of the Detrended Correspondence Analysis (DCA) of fossil pollen data from Lake Acopia. Local pollen zones are color coded, and samples that contained more or less than 2% *Sporormiella* spores (spores were excluded from the analysis) are marked by shape.

**Figure 9 plants-13-01019-f009:**
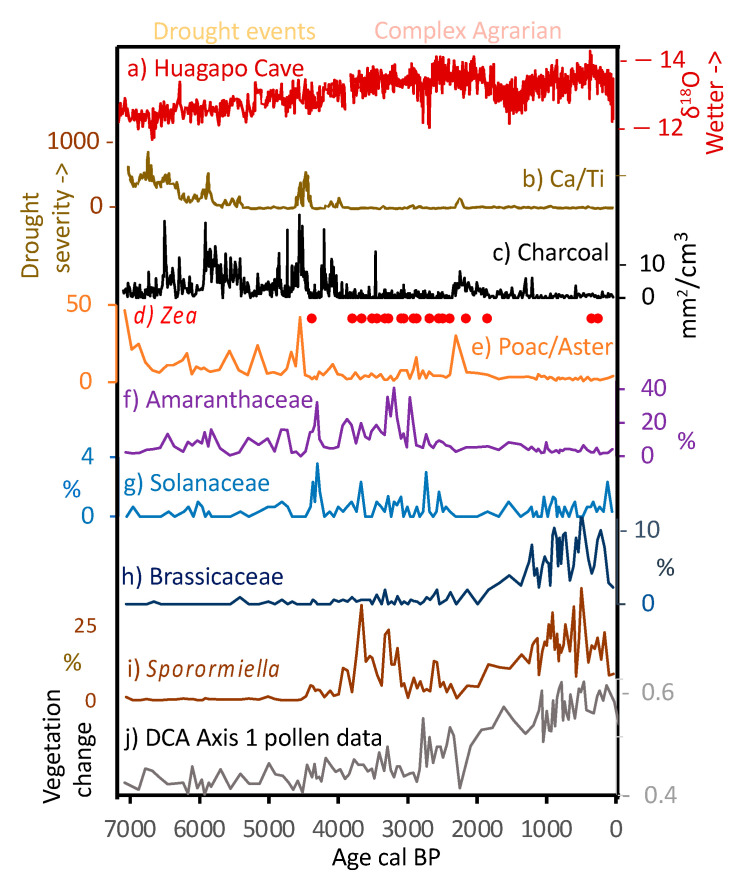
Evidence of climate change, cultivation, and vegetation trajectories from Lake Acopia, Peru, relative to a regional climate history. Isotopic data (a) are from Huagapo Cave [56], all other data are from Lake Acopia. Gold bars highlight local drought events defined by high Ca/Ti ratios. Bold line on b represents a 20-yr running mean, with full resolution data as paler line.

**Figure 10 plants-13-01019-f010:**
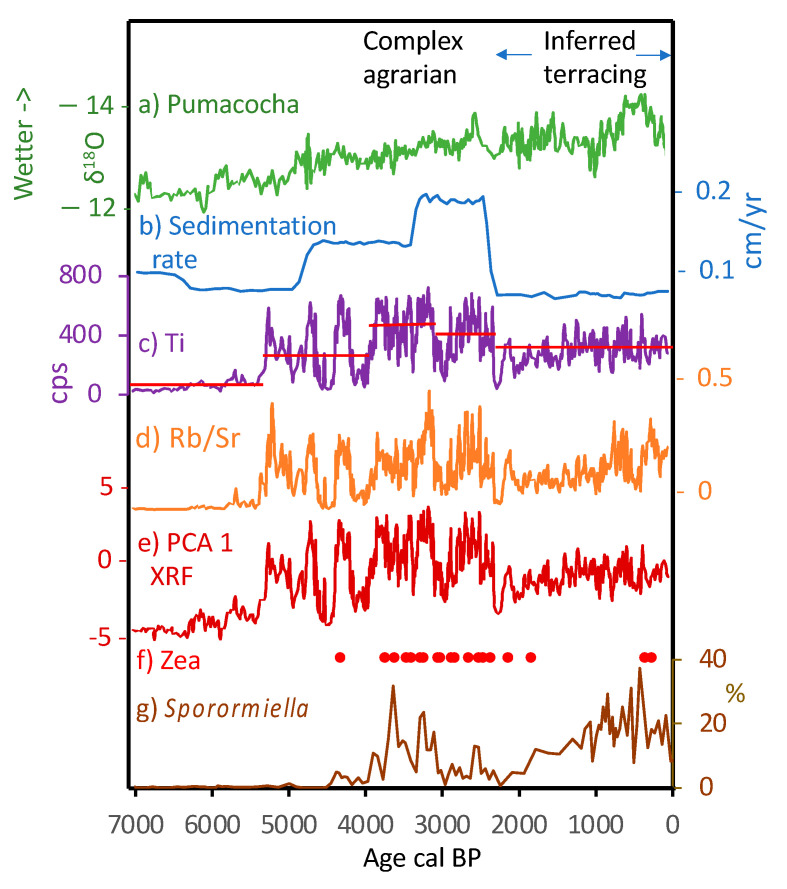
Multiproxy data from Lake Acopia showing changes in land use and erosion in relation to regional climate and inferred terracing. Isotopic data (a) from Lake Pumacocha [55]. Bold line on c, d, and e represents 20-yr running mean, with full resolution data as paler line. Red horizontal lines show mean value of Ti input during five periods, defined by Bayesian changepoint analysis within the record.

**Figure 11 plants-13-01019-f011:**
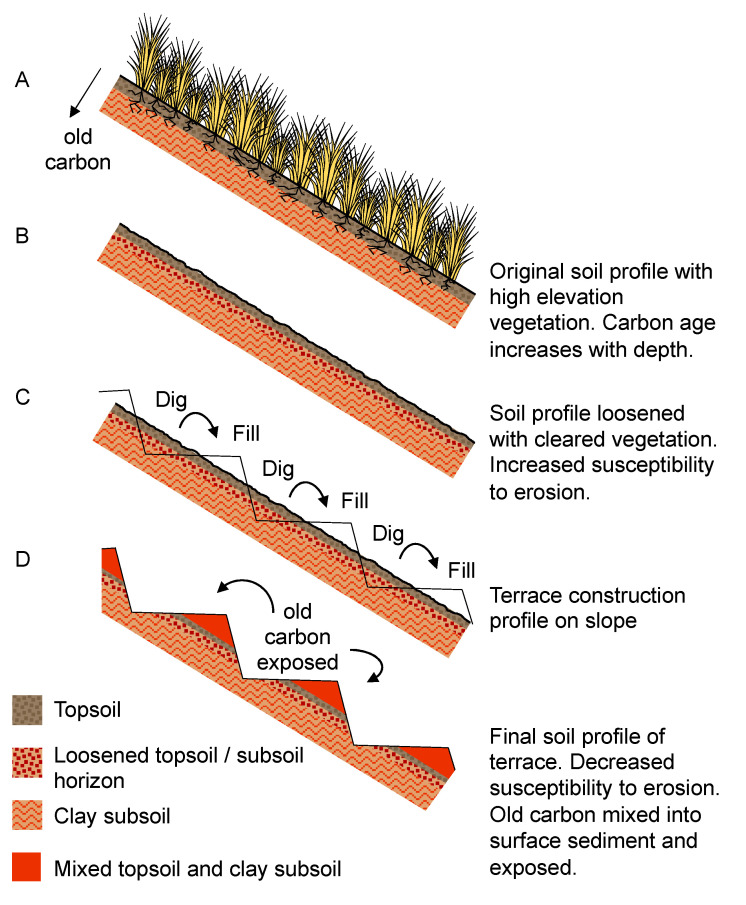
Hypothesized influence of terracing upon Andean landscapes and old carbon input.

**Table 1 plants-13-01019-t001:** Results of AMS dating of eight bulk sediment samples from Lake Acopia, Peru. Ages were calibrated using rbacon [52] and the IntCal20 calibration curve [40].

Lab I	Depth (cm)	^14^C (yr BP) ± 1 σ			cal BP 2σ Range			Median Age Cal BP	^13^C
OS-135406 *	7	5070	±	40	5660	–	5897	5817	−27.51
OS-125490 *	71	4500	±	20	5159	–	5287	5167	−25.5
DAMS-052062 *	79	3814	±	34	4089	–	4396	4203	
OS-126479 *	81	2610	±	20	2698	–	2757	2749	−24.35
OS-135405 *	120	3140	±	25	3222	–	3383	3368	−27.23
OS-125489	156	2480	±	20	2633	–	2701	2589	−25.4
OS-125503	365.5	3070	±	20	3283	–	3345	3291	−25.84
OS-125504	578	4180	±	20	4570	–	4823	4728	−25.91
OS-125505	698	5610	±	25	6298	–	6407	6372	−26.69

* Indicates rejected radiocarbon date.

**Table 2 plants-13-01019-t002:** The timing of peak abundances of Brassicaceae and Solanaceae pollen in other Tropical Andean fossil pollen records. Sites are sorted be decreasing elevation. Yellow highlighting denotes the sites at an elevation where potatoes or maca could potentially be grown.

	Max % of Brassicaceae%	Peak Time Cal BP	Max % Solanaceae%	Peak Time Cal BP	Site Elevation (masl)	Source
Cerro Llamoca	11	600	2	400	4450	[65]
Caserococha	3	3000	3	2000	3900	[66]
Titicaca	3	4000	0.6	4720	3810	[53]
Acopia	11.3	424	3.5	4280	3750	this ms
Huaypo	3.7	340	0.4		3500	[63]
Natosa Peat bog	5	4000	<2		3482	[67]
Challacaba	7	300	<2		3400	[68]
Marcacocha	30	−20	<2		3350	[62]
Chochos	1		1.9		3285	[69]
Pacucha	6	270	1.5		3100	[54]

## Data Availability

The datasets generated from this study are available through GITHUB which include 14C, pollen, charcoal, loss-on-ignition (carbonate) and XRF (Ti, Si, and Ca).

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
