# Peer review of "The Evolution of Agrarian Landscapes in the Tropical Andes"

_plants, 2024, doi:10.3390/plants13071019_

Round 1
Reviewer 1 Report
Comments and Suggestions for Authors
The study presents a high-resolution paleoecological analysis of a 7000-year history of changing climate and land management around Lake Acopia in the Andes of southern Peru. The study explores the complex interaction between climate change and human land-use practices, aiming to understand processes of adaptation and change over time.
While the study provides valuable insights into the role of humans in shaping Andean landscapes and the incremental nature of land-use practices, it is based on a single case study. The findings may not represent broader patterns or apply to other regions with different ecological and cultural contexts. It would be beneficial to conduct similar studies in different locations to assess the generalizability of the observed trends and processes.
The interpretation of the data should be cautious, and alternative explanations or potential biases in the analysis, specifically in the discussion section, should be considered.
The study focuses on the past and does not explicitly discuss the relevance of the findings for current or future land-use practices in the region.
Introduction should be divided into different paragraphs. Like tropical Andes its significance, and historical background and recent developments.
Agrarian landscapes its potential for agriculture and recent advances. In the third paragraph discuss the process of evolution in Tropical Andes, its impacts and agriculture practices.
Line 53-54 cite recent study as well doi: https://doi.org/10.1016/j.iswcr.2023.07.002
Considering the ongoing and projected climate change, it would be valuable to discuss the implications of the study's findings for informing sustainable land-use strategies and adaptation measures in the face of changing environmental conditions in the conclusion section.
Author Response
We thank the reviewers for their insights and critique. Below are responses (in blue) to all the points raised.
Reviewer #1
The study presents a high-resolution paleoecological analysis of a 7000-year history of changing climate and land management around Lake Acopia in the Andes of southern Peru. The study explores the complex interaction between climate change and human land-use practices, aiming to understand processes of adaptation and change over time.
While the study provides valuable insights into the role of humans in shaping Andean landscapes and the incremental nature of land-use practices, it is based on a single case study. The findings may not represent broader patterns or apply to other regions with different ecological and cultural contexts. It would be beneficial to conduct similar studies in different locations to assess the generalizability of the observed trends and processes.
We do not disagree that additional studies would be beneficial. This is the first study encompassing this age range, detail, and integration of archaeology and paleoecology. Other studies can test our hypotheses.
The interpretation of the data should be cautious, and alternative explanations or potential biases in the analysis, specifically in the discussion section, should be considered.
We believe that we have been cautious and put forward the best explanation of a complex dataset. We do not offer alternative explanations, partly to keep the document readable, but primarily because we do not believe them to be likely.
The study focuses on the past and does not explicitly discuss the relevance of the findings for current or future land-use practices in the region.
Yes, this is a paleoecological study and comments on modern policy etc are not within the scope of this study.
Introduction should be divided into different paragraphs. Like tropical Andes its significance, and historical background and recent developments.
Agrarian landscapes its potential for agriculture and recent advances. In the third paragraph discuss the process of evolution in Tropical Andes, its impacts and agriculture practices.
We agree that the introduction was indigestible and have subdivided it into:
An historical overview of early settlement and cultivation; and
Human responses to Holocene climate change and the temporal scale of study.
Line 53-54 cite recent study as well doi: https://doi.org/10.1016/j.iswcr.2023.07.002
This study does not seem to be directly relevant to our analysis
Considering the ongoing and projected climate change, it would be valuable to discuss the implications of the study's findings for informing sustainable land-use strategies and adaptation measures in the face of changing environmental conditions in the conclusion section.
That is beyond our expertise and the scope of this article.
Reviewer 2 Report
Comments and Suggestions for Authors
Dear Authors!
In my opinion this work is based on a very valuable research and contain a lot of novel information about the evolution of agricultural landscapes in Andes mountain. And also that work give a good example about how we can use palynological analyses, radiocarbon age dating and the results of elemental composition to support a hypothesis about landscape evolution.
There are some minor questions and remarks insert in the text, because some thing in the methodological part was not completelly clear for me.

Author Response
In my opinion this work is based on a very valuable research and contain a lot of novel information about the evolution of agricultural landscapes in Andes mountain. And also that work give a good example about how we can use palynological analyses, radiocarbon age dating and the results of elemental composition to support a hypothesis about landscape evolution.
Thank you
There are some minor questions and remarks insert in the text, because some thing in the methodological part was not completelly clear for me.
Why did You choose Lake Acopia? Why this lake important (or special) from the aspect of agrarian landscape evolution?
We have added the following to the site description “This lake lies within the region of the Wari (c. AD 600 – 1000; 1350-950 cal BP) and Inka (c. AD 1400-1533; 550- 417 cal BP) empires and is likely to have been occupied for a much longer period. The pre-Wari history of this region is poorly known, and yet it is considered to lie within the cradle of plant and animal domestication (Pearsall 2008)”
What is the sedimentation rate in the lake? How thick a layer is deposited on the bottom of the lake? If the sedimentation is fast, the bathymetrical data of the year of 2008 is not relevant in the year of 2023.
Reviewer found their own answer to this. Rates are slow and will not impact bathymetry.
The sampling method not completely clear for me. How many samples do collect? Where do You collect from the samples from? From the lake bed? Or do You drive a core below the river bed? What was the depth of Your sediment coring? that 7.74 m value?
We have clarified where the core was collected and inserted an additional call-out to the figure that shows the coring location.
What kind of material did You measure radiocarbon data from? Organic matter, remnants of lake vegetation or something other?
All 14C ages are derived from bulk carbon as there were no macrofossils to date in these sediments.
Maize was the only producted vegetation by ancient peoples or there were any agricultural vegetation too?
We describe finding the pollen of potatoes, maca, and quinoa all of which are ancient crops.
Why was charcoal the proxie of human landscape disturbance? In my opinion, charcoal could be get/ washed into a lake in a natural way: for example when a mass movement triggered on the shore of the lake, a mass movement can transport the shrubs/trees into the lake. Or Do You use that kind of charcoals which origin from agricultural/planned vegetation?
Thank you for pointing this out: We have added the following sentence. “As regular fire in the Holocene Andes was so rare before human arrival (Schiferl et al. 2023), its regular presence was strongly linked to human activity (White 2013, Bush et al. 2022).”
How did You differentiate ACP-1, ACP-2, ACP-3 fom each other? On the basis of sedimentation rate?
We have added the following sentences: “The local pollen zones were defined by a CONISS analysis – a multivariate tool that identified the strongest ecological divisions the a time-series.Three local zones were significant using a broken-stick model.” We have added a clarification in methods and results.
Did You find maybe any soot particles in pollen spectrum which also would be the evidence of fire events?
No, soot is not usually used as a marker as it is potentially transported very long distances and is not as informative as charcoal regarding the presence of fire in the adjacent watershed.
Reviewer 3 Report
Comments and Suggestions for Authors
This manuscript inferred the process of land use change in Lake Acopia in the Andes of southern Peru through the study of pollen fossil content, and comprehensively considered natural conditions and human activities, summed up the formation process of agricultural landscape in this region, and provided new insights into the role of human influence. However, there are still many problems in this article, please check it carefully and make corrections. The comment is Major Revision.
Line 22,39,41,50,83,178,196,221,230,240,250,266,283,292,307,326,368,394,411,463,523:Please add separators for numbers over 1000, and please carefully check the relevant numbers in the article and make corrections, including the numbers in the figures and tables.
Topic: There is too much difference between the scope of the topic and the actual research area, so you can consider being more specific and narrow the scope of the research in the topic.
Line 84-85:The data of average annual precipitation and average annual temperature here are the data of 2013. With the global warming, the temperature in some parts of the tropics has increased, and the precipitation has also increased, the data may be adjusted, please find the latest reference literature, the data can be more accurate.
Line 82-93:The paragraph format here is different from the context. Is there an error? If there is a mistake, please correct it.
Line 109-110:How does this heat up? What kind of equipment? Please add here.
Line 128-134:Consider attaching some pictures of experimental instruments to facilitate readers' better understanding.
Line 149-152:The paragraph format here is different from the context. Is there an error? If there is a mistake, please correct it.
Figure 7:The picture clarity is not particularly good, and the color in the picture is not too clear, I hope to improve it.
Line 491-492:The font format is inconsistent with the context. Please correct it.
References:The references are too old, large but old, please refer to the latest research results,such as (https://doi.org/10.1016/j.watres.2022.119065).
Comments on the Quality of English Language
This manuscript inferred the process of land use change in Lake Acopia in the Andes of southern Peru through the study of pollen fossil content, and comprehensively considered natural conditions and human activities, summed up the formation process of agricultural landscape in this region, and provided new insights into the role of human influence. However, there are still many problems in this article, please check it carefully and make corrections. The comment is Major Revision.
Line 22,39,41,50,83,178,196,221,230,240,250,266,283,292,307,326,368,394,411,463,523:Please add separators for numbers over 1000, and please carefully check the relevant numbers in the article and make corrections, including the numbers in the figures and tables.
Topic: There is too much difference between the scope of the topic and the actual research area, so you can consider being more specific and narrow the scope of the research in the topic.
Line 84-85:The data of average annual precipitation and average annual temperature here are the data of 2013. With the global warming, the temperature in some parts of the tropics has increased, and the precipitation has also increased, the data may be adjusted, please find the latest reference literature, the data can be more accurate.
Line 82-93:The paragraph format here is different from the context. Is there an error? If there is a mistake, please correct it.
Line 109-110:How does this heat up? What kind of equipment? Please add here.
Line 128-134:Consider attaching some pictures of experimental instruments to facilitate readers' better understanding.
Line 149-152:The paragraph format here is different from the context. Is there an error? If there is a mistake, please correct it.
Figure 7:The picture clarity is not particularly good, and the color in the picture is not too clear, I hope to improve it.
Line 491-492:The font format is inconsistent with the context. Please correct it.
References:The references are too old, large but old, please refer to the latest research results,such as (https://doi.org/10.1016/j.watres.2022.119065).
Author Response
This manuscript inferred the process of land use change in Lake Acopia in the Andes of southern Peru through the study of pollen fossil content, and comprehensively considered natural conditions and human activities, summed up the formation process of agricultural landscape in this region, and provided new insights into the role of human influence. However, there are still many problems in this article, please check it carefully and make corrections. The comment is Major Revision.
Line 22,39,41,50,83,178,196,221,230,240,250,266,283,292,307,326,368,394,411,463,523:Please add separators for numbers over 1000, and please carefully check the relevant numbers in the article and make corrections, including the numbers in the figures and tables.
We are not doing this as in standard English (US and UK) such a separator is only used for numbers exceeding 9999.
Topic: There is too much difference between the scope of the topic and the actual research area, so you can consider being more specific and narrow the scope of the research in the topic.
Sorry, I don’t understand this comment. We do not have a topic stated in this manuscript.
Line 84-85:The data of average annual precipitation and average annual temperature here are the data of 2013. With the global warming, the temperature in some parts of the tropics has increased, and the precipitation has also increased, the data may be adjusted, please find the latest reference literature, the data can be more accurate.
We have updated the climate data, Cusco being the nearest city, and cited the source (https://en.climate-data.org/south-america/peru/cusco/cusco-1016/#climate-graph; accessed 2.26.2024).
Line 82-93:The paragraph format here is different from the context. Is there an error? If there is a mistake, please correct it.
Sorry, I do not see the specific issue. Line 82-93 is within a paragraph, which I agree needs to be broken up. I hope our revisions are acceptable.
Line 109-110:How does this heat up? What kind of equipment? Please add here.
We have added “heated in a muffle furnace”.
Line 128-134:Consider attaching some pictures of experimental instruments to facilitate readers' better understanding.
That is not normally done for such standard techniques.
Line 149-152:The paragraph format here is different from the context. Is there an error? If there is a mistake, please correct it.
Our line-numbers are not lining up, but I don’t see an issue.
Figure 7:The picture clarity is not particularly good, and the color in the picture is not too clear, I hope to improve it.
I think this may be a problem in the pdf generation the figure in text in the word document is fine, but we will be sur to verify this in later versions.
Line 491-492:The font format is inconsistent with the context. Please correct it.
Sorry, again I do not see any issue. All text is in Calibri font. Perhaps again it is the bold font of the figure caption?
References:The references are too old, large but old, please refer to the latest research results,such as (https://doi.org/10.1016/j.watres.2022.119065).
It is important to cite foundational literature, and relevance is more important than the very latest publication date. Naturally we will look to see if there are any 2023/2024 publications that we think should be added.